# Robust-DSN: A Hybrid Distributed Replication and Encoding Network Grouped with a Distributed Swarm Workflow Scheduler

Zeeshan Hameed [1,*], Hamid R. Barzegar [1], Nabil El Ioini [2] and Claus Pahl [1]

1   Faculty of Engineering, Free University of Bozen-Bolzano, 39100 Bozen-Bolzano, Italy;
    hamidreza.barzegar@unibz.it (H.R.B.); claus.pahl@unibz.it (C.P.)
2   Faculty of Computer Science, University of Nottingham, Semenyih 43500, Selangor, Malaysia;
    elioini.nabil@nottingham.edu.my
*   Correspondence: zeeshan.hameed@student.unibz.it

**Abstract:** In many distributed applications such as the Internet of Things (IoT), large amounts of data are being generated that require robust storage solutions. Traditional cloud solutions, although efficient, often lack trust and transparency because of centralized management. To address these issues, we present Robust-DSN, a distributed storage network leveraging the hybrid distributed replication and encoding network (HYDREN) and the distributed swarm workflow scheduler (DSWS) as its main components. Our system uses an interplanetary file system (IPFS) as an underlay storage network and segments it into multiple regions to distribute the failure domain and improve the data's proximity to users. HYDREN incorporates Reed–Solomon encoding and distributed replication to improve file availability, while DSWS optimizes resource allocation across the network. The uploaded file is encoded into chunks and distributed across distinct optimal nodes leveraging lightweight multithreading. Additionally, Robust-DSN verifies the integrity of all chunks by preserving the hashes when uploading and validating each chunk while downloading. The proposed system provides a comprehensive solution for resilient distributed data storage, focusing on the key challenges of data availability, integrity, and performance. The results reveal that compared with a state-of-the-art system, the proposed system improves file recovery by 15%, even with a 50% peer failure rate. Furthermore, with replication factor 4 and the same failure resilience as IPFS, it saves 50% storage and enhances file recovery by 8%. Robust-DSN acts as a distributed storage platform for emerging technologies, expanding storage system capabilities in a wide range of distributed applications.

**Keywords:** distributed storage system; file availability; swarm intelligence; distributed replication; interplanetary file system (IPFS)





## 1. Introduction

There is an exponential growth of data generated at the network edges due to the growth of Internet of Things (IoT) devices and mobile applications. Cloud storage is considered to be a potential solution for managing these data. Cloud storage solutions provide a centralized remote system established through the cooperation of multiple networked devices [1]. In a cloud storage system, a service provider is responsible for managing and securing the user's confidential data [2,3]. From the user's point of view, a single entity for data management may also be a single point of failure [4,5]. In addition, questions of trust and transparency about how securely confidential user data are managed and stored arise.

As an alternative to cloud services, distributed storage systems have emerged [6]. In a decentralized storage system, multiple servers work together to provide desired services to users. Distributed systems follow P2P networking, where every node in the network has the same rights to provide the services [7]. Since the data are stored in multiple places, these systems have the potential to be more robust against large-scale outages and possible

censorship. However, the decentralized features of these storage systems introduce various challenges [8]. The major concern is that the collaborators are unreliable and untrusted. To achieve adequate guaranteed data availability, the distributed storage systems must apply a high level of redundancy to the content to ensure data availability when requested. Moreover, data may be lost if a collaborator loses or leaves its link to the network.

To improve the level of data availability, distributed storage systems usually have two independent primary approaches: coding-based techniques and data replication [9,10]. The replication approach is quick and simple; however, the system has to bear a large storage overhead. In contrast, coding-based approaches are computationally complex and may raise the issue of large computation overhead, resulting in degraded system performance if used inappropriately. Moreover, the heterogeneous nature of storage nodes in a distributed environment makes the process more complex for reliable data persistence and fault tolerance. There should be an appropriate approach to analyze the inherent trade-off between performance and storage overhead for effective utilization of the system resources [11,12]. Therefore, a comprehensive solution needs to improve both aspects at the same time in distributed storage systems.

Another challenge is related to task scheduling. Searching for suitable nodes for storing chunks of data from a pool of diverse and heterogeneous storage nodes is a critical challenge. This problem includes several concerns, including storage size, performance capabilities, and availability to provide services. Another crucial function is associated with preserving the integrity of each of the data chunks. The data would not be reconstructed even if a single data chunk was tampered with by any of the storage nodes [13,14]. Moreover, the issues related to the processing time of service requests are also crucial. Storage systems might be slow, specifically if the data are not broadly replicated or because of network congestion between collaborators storing the required data. Distribution and fetching of data chunks would hence be executed concurrently to minimize the data retrieval time for a user request. Thus, the algorithms of the overlay network and backend abstraction layers should be synchronized to maximize concurrent work while minimizing recovery time. Consequently, a system that collectively combines fault tolerance, data integrity, and identification of the most suitable node for task placement and that provides an efficient method for transferring and retrieving chunks to and from a distributed storage system remains a research challenge.

To address these availability, integrity, performance, and scheduling challenges in distributed storage networks, we present the robust distributed storage network (Robust-DSN), which aims at improving data availability, integrity, task scheduling, and execution time simultaneously. Our proposed solution creates a logical network on top of the physical storage network and acts as an additional layer between the user and the distributed storage network. We use the interplanetary file system (IPFS) as an underlay network for content-based data management. Robust-DSN integrates a novel hybrid distributed replication and encoding network (HYDREN) and distributed swarm workflow scheduler (DSWS). Our contributions are summarized as follows: (1) constructing a HYDREN for improving file availability, (2) designing a distributed optimizer (DSWS) for scheduling the storage tasks effectively, (3) applying the message digest (md5) algorithm to each of the data segments to maintain integrity, and (4) constructing a lightweight multithreading approach for handling the chunk distribution and file reconstruction to improve overall task execution time.

Our results prove that data availability and integrity for decentralized storage systems can be achieved without substantial storage overhead. Our evaluations show that the proposed system is capable of improving file availability and integrity, storage utilization, and task execution performance simultaneously. Specifically, with the configuration of four replication factors and the same failure resilience as IPFS, the proposed system uses 50% less storage and offers 8% more file recovery likelihood. Compared with state-of-the-art systems, our proposed system offers 15% more file recovery likelihood, even for a peer failure rate of 50%.

## 2. Related Work

The growth of distributed storage systems has been significant with the evolution of data generation. Modern applications demand distributed storage solutions that are not only fault-tolerant and storage-efficient but also efficient in terms of performance and data integrity simultaneously. Researchers have proposed various techniques to deal with data reliability [15]. To ensure data reliability, the replication strategy is the common redundancy mechanism used for distributed storage systems. For instance, the Google file system (GFS) forms three replicas of each of the data chunks and guarantees that all replicas of a chunk are not stored on a similar rack. Although this approach ensures a high level of data availability, a storage space of $M + 1$ times is required to deal with $M$ failures. This produces massive storage overhead. For instance, by creating three copies to tolerate the failure of two data chunks, the rate of storage unitization of the system would only be 33.3%.

Recently, erasure coding has received significant attention and emerged as a promising alternative to replication for data storage in distributed storage systems [16]. In this method, the original data are split into data chunks ($d$) and then encoded to generate additional parity chunks ($p$). The benefit of erasure coding is its capability to provide comparable data reliability with less storage space than the traditional replication approach. A prominent application of this approach can be seen in Facebook's $F4$ storage system, which applies erasure coding with the parameters ($p = 4, d = 10$). This method not only improves system resilience but also reduces storage space by about 1.6 times compared to the replication approach. However, erasure coding also faces some challenges. It introduces the substantial computational overhead of the data encoding and decoding. Furthermore, with the coding parameters ($d = 10, p = 4$), at least 10 out of 14 untempered chunks are required to reconstruct the data.

Understanding the importance of distributed storage systems, some special overlay networks (e.g., IPFS, Swarm) have been developed that incorporate necessary protocols and functions for building customized applications on top of them [6,17]. The nodes in these systems are connected in a peer-to-peer topology to manage version control and have been conventionally used as a storage layer for various applications. Implementing IPFS and Swarm with blockchain has been an effective approach to improving system functionalities in various aspects, such as access control and data integrity [18,19]. This integration has been employed in various domains, including healthcare, agricultural logistics, file management systems, and the Internet of Things (IoT) [20]. Moreover, these networks are also effective without blockchain since they have been featured for unique applications, for instance, to archive web content and to assist with edge and fog computing storage requirements [21]. However, storage nodes in Swarm and IPFS are heterogeneous, and the data are replicated based on the popularity of the data within the network. They do not guarantee data persistence and fault tolerance as a built-in functionality. BitTorrent is a file-sharing system that uses peer-to-peer (P2P) network topology [22]. BitTorrent protocol splits larger files into smaller chunks and then shares them between the users. It allows users to upload and download files simultaneously; while downloading a file, the BitTorrent client helps to find other users who have the chunks of that file. The client can download the chunks from different peers. Moreover, when the client receives the file, it also starts uploading chunks to serve these chunks to other users. This is optional and depends upon the clients and whether they want to be a part of distributing the downloaded file (act as seeders). However, BitTorrent heavily relies on seeders. If a file has limited seeders, the downloading speeds can be very slow, and a file may even become unavailable [23]. Moreover, there is no built-in mechanism for data mutability, and users need to rely on third-party solutions to guarantee the integrity of the data. Therefore, BitTorrent might have similar issues to IPFS, as for both systems, it is impossible to change or delete content once distributed.

To address the inherent constraints of the IPFS framework, specifically regarding data persistence and redundancy, Filecoin is building an incentive layer over IPFS [24]. Adding

an incentive layer, Filecoin incentivizes service providers with cryptocurrency tokens for storing and duplicating the content, thereby improving content availability. Regardless of its benefits, Filecoin's replication methodology increases the associated storage costs, network bandwidth, and latency.

There is a decentralized storage system called Arweave influenced by blockchain technology. Arweave is a decentralized data storage platform that has a blockchain-based structure called blockweave [25,26]. Its design primarily focuses on immutable long-term data storage solutions. Arweave ensures that once a file is uploaded, it remains accessible forever. Each piece of data is considered as a block that is linked to one or more previous blocks to construct a chain, as blockchain does. The data retrieval process in Arweave is performed via traditional hypertext transfer protocol (HTTP) GET requests using Arweave gateways. However, the immutability of data on Arweave raises concerns regarding data administration. Moreover, since it is an on-chain data storage platform, the scalability of its structure may cause some issues; for example, the continuous growth of the blockweave may lead to an increase in the operational burden and possibly slower data retrieval time [27].

Hypercore is another group of protocols devised for building peer-to-peer (P2P) applications including distributed file systems similar to BitTorrent. The Hypercore network consists of different storage modes, where nodes decide which data of a directory and version they want to store. The protocol can be assumed as a shared folder where the files can be added, modified, and deleted. Since communication between peers is encrypted using asymmetric encryption, it is compulsory to have a specific read key to look up and read the data. However, in Hypercore, the data accessibility completely depends upon the existence of peers storing the data. If a peer in the network is the only peer hosting crucial data and it goes offline, the data would become inaccessible. Moreover, Hypercore does not use content-based addressing; therefore, it relies on additional solutions to verify the integrity of the data.

The presented study here also addresses the limitations of our previous work on the improvement in file availability and performance of DSNs [28]. In our previous work, several challenges were not sufficiently addressed. The file availability was improved based on the encoding mechanism only, where the probability of the data loss can be expressed in a restricted format (see Equation (6) below). Moreover, the PSO algorithm was designed in a limited form, i.e., being executed in a single region at a time, and particles were only evaluated in sequence. Furthermore, we provide here a broader range of solution-specific and comparative evaluations.

Most existing distributed systems have not adequately addressed the trade-off between improving data availability, storage overhead, and integrity persistence. Moreover, most of the systems that improve file availability use a replication strategy and replicate a full copy of the file to multiple nodes instead of splitting a file into multiple chunks and ideally distributing them to distinct storage nodes. The level of file distribution depends upon the number of copies present in the network. Moreover, the challenge of effectively scheduling the workflows and finding the most suitable service providers for executing them from a diverse group with heterogeneous resources is a critical challenge that demands greater concentration for the development of distributed systems. Therefore, a significant research gap regarding the evolution of methodologies remains that is required to improve file availability and integrity, identify optimal nodes for data storage, and improve the overall system performance. The comparison between proposed and various decentralized storage systems is shown in Table 1.

**Table 1.** Comparison of various storage systems.

| Properties\Systems | BitTorrent | IPFS | Swarm | Hypercore | Arweave | Filecoin | Proposed |
|---|---|---|---|---|---|---|---|
| Each node stores | Entire file | Entire file | Entire file | Entire file | Entire file | Entire file | **Chunk** |
| Data persistence | No | No | No | No | Yes | Yes | **Yes** |
| Immutability | No | Yes | Yes | No | Yes | Yes | **Yes** |
| Load balancing | No | No | No | No | No | No | **Yes** |
| Integrity (content address-based) | No | Yes | Yes | No | blockweave | Yes | **Yes** |
| Allow modification and deletion | No | No | No | Yes | No | No | **Yes** |
| Data availability (encoding-based) | No | No | No | No | No | No | **Yes** |
| Data availability (replication-based) | Yes | Yes | Yes | Yes | Yes | Yes | **Yes** |
| Storage location | Random | Random | Random | Random | Random | Random | **Find best nodes** |

## 3. Robust-Dsn System Design

Our complete solution, called Robust-DSN, consists of distinct solutions for distributed replication and encoding, distributed swarm scheduling, and processes for the upload and download of data. The overall system architecture and the distinct solution components will now be introduced.

### 3.1. Architecture and Components

Our primary objective is to improve data availability, persist data integrity, and identify optimal nodes for data storage in a cryptographic distributed storage network with as minimal as possible storage overhead and processing time. Robust-DSN allows users to control the resilience level and enables file recovery despite a large-scale failure of storage nodes happening in the underlying storage network. Robust-DSN uses IPFS as an underlay storage layer and requires no modification to the underlying storage system. The system architecture of the Robust-DSN system is shown in Figure 1. This system architecture is separated into three layers: DSN users, DSN infrastructure itself, and DSN service providers. Figure 2 describes the underlay storage network where $H$ is the home region and $H_1, H_{-1}, ... H_z$ is the chain of neighboring regions. Joining the DSN network as a service provider is simple, requiring the installation of an IPFS node to connect to the network. For our experiments, a virtual machine (with resources of 16 GB RAM, 256 GB of storage, and a five-core CPU) hosting multiple Docker containers was set up to mimic a private cluster of DSN service providers. Each container was allocated different resources to indicate a heterogeneous resource environment. The users request on-demand and reliable storage selections from the service provider network, delivering their data along with corresponding performance requirements. As the main components, the DSN infrastructure incorporates a so-called HYDREN algorithm that ensures data reliability and integrity, the DSWS component to search the most appropriate service providers to ensure performance requirements, a REST API, and a mechanism for processing the requests concurrently.

### 3.2. Distributed Replication and Encoding Network

HYDREN (hybrid distributed replication and encoding network) consists of the following major components: an encoder, a decoder, a replicator, and a message-digest (md5) algorithm. In our distributed storage system, we split our underlay storage network into multiple geographical areas. The purpose of this approach is to save the chunks of a file as close as possible to the users. The chunks of a file will only be stored in other regions when a region does not have the required resources to provide services. Moreover, this approach distributes the failure domain of the storage network by distributing the chunks to the neighboring region and replicating them to the global best nodes searched by the DSWS algorithm to further enhance file availability. Thus, HYDREN begins by taking an uploaded file and processing it using Reed–Solomon encoding. The encoding process splits files into multiple data chunks, and using the data chunks $d$, it calculates and adds parity chunks $p$. After that, to ensure the integrity of each of the chunks, the message-digest (md5) algorithm is applied to calculate the hash of each of the chunks before distributing.

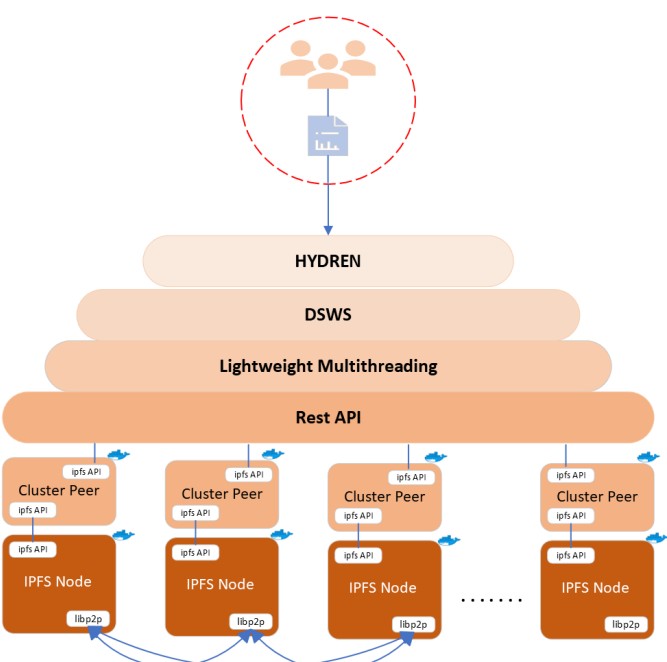

**Figure 1.** Overall Robust-DSN system architecture.

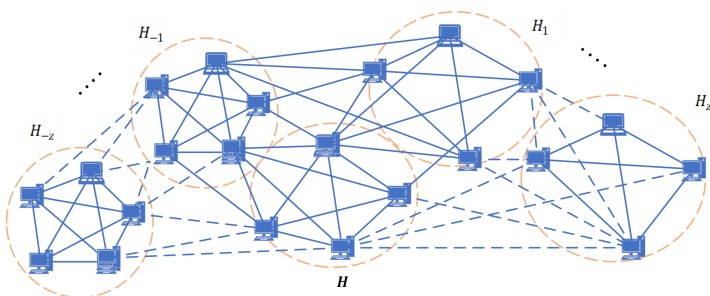

**Figure 2.** Schematic network topology.

md5 is chosen here as a compromise between the performance and strength of the security provided—the selection of other options would rebalance the two qualities.

The detailed formalization of encoding and decoding will now be explained. We can consider a polynomial $f(x)$ of degree $n$ having $n+1$ coefficients.

$$f(x) = c_0 + c_1 x + c_2 x^2 + \ldots + c_n x^n \tag{1}$$

$$f(x_0) = a_0 + a_1 x_0 + a_2 x_0^2 + \cdots + a_n x_0^n = y_0$$
$$f(x_1) = a_0 + a_1 x_1 + a_2 x_1^2 + \cdots + a_n x_1^n = y_1$$
$$\vdots$$
$$f(x_n) = a_0 + a_1 x_n + a_2 x_n^2 + \cdots + a_n x_n^n = y_n$$

$$\underbrace{\begin{bmatrix} 1 & x_0 & x_0^2 & \cdots & x_0^n \\ 1 & x_1 & x_1^2 & \cdots & x_1^n \\ \vdots & \vdots & \ddots & \vdots & \vdots \\ 1 & x_n & x_n^2 & \cdots & x_n^n \end{bmatrix}}_{\text{Vandermonde matrix (V)}} \begin{bmatrix} a_0 \\ a_1 \\ \vdots \\ a_n \end{bmatrix} = \begin{bmatrix} y_0 \\ y_1 \\ \vdots \\ y_n \end{bmatrix}$$

One of the important characteristics of the polynomial is that a polynomial of degree $n - 1$ can be uniquely characterized by any $n$ points that lie on that polynomial function. In other words, for any $n$ distinct values of $x$, we can derive the $n + 1$ coefficients $c_0$ to $c_n$, and these coefficients will remain the same, making no difference in which n values of variable $x$ we use. Since we need only n points to define the $n - 1$ polynomial, the points $n + 1, n + 2. \ldots$, and so on are redundant. After finding the coefficients, we can evaluate the polynomial with as many values of $x$ as we want. Those extra points are redundant and are not essential to define the polynomial. This redundant information allows us to find the original values in case the data are corrupted during transmission. Therefore, we performed the following operations to design an effective encoder: (1) split the data into $d$ chunks, (2) set the data chunks as the values of the polynomial $p(x)$ at values of $x$; we consider it as the order of the data chunk, (3) determine the coefficients of this polynomial $p(x)$ from the $(x_i, d_i)$ pairs, (4) evaluate polynomial $p(x)$ for an additional $p$ values of $x$ to create parity chunks, and (5) calculate the hash of all $n(d + p)$ chunks. This metadata related to a particular file is preserved for successfully decoding the file. Hashes of the chunks and the order of the chunks help us to authenticate and locate the corrupted chunks. When we receive the chunks, we calculate the hash of the chunks. Since hash is a unique identifier of that particular chunk, we can identify the order associated with that chunk. If we lose the data chunk, any d rows of the Vandermonde matrix can be taken to make a $d \times d$ matrix, which will be invertible and used to generate lost chunks.

The encoding process converts original data into coded data that consist of both data and parity chunks by applying a generator matrix $G$ [29,30]. The generator matrix is a Vandermonde matrix that is used to perform matrix operations on data chunks for generating parity chunks [31]. The property of the Vandermonde matrix is that any subset of rows that forms a square matrix guarantees an invertible matrix. For an encoding with $d$ data chunks and $p$ parity chunks (total $n = d + p$), the generator matrix would be an $n \times d$ matrix. It is structured as follows, where $x_i$ are nonzero elements of a finite field (generally $GF(2^8)$):

$$G = \begin{pmatrix} 1 & x_1 & x_1^2 & \cdots & x_1^{d-1} \\ 1 & x_2 & x_2^2 & \cdots & x_2^{d-1} \\ \vdots & \vdots & \vdots & \ddots & \vdots \\ 1 & x_n & x_n^2 & \cdots & x_n^{d-1} \end{pmatrix}$$

For the data chunks $D = [D_1, D_2, \ldots, D_d]$, the encoding process would be a matrix multiplication:

$$C = G \cdot D \tag{2}$$

The resulting vector $C$ contains both the original data and the parity chunks. If some of the chunks are lost or tampered with, we can retrieve the original data by employing any $d$ out of the $n = d + p$ chunks. A subset of the encoded vector $C'$ and the associated rows of the generator matrix $G$ would be used to generate $C'$ (denoted as $G'$); the decoding process consists of solving the following linear equation:

$$C' = G' \cdot D \tag{3}$$

Since we need $d$ chunks to reconstruct the file, $G'$ would be a square matrix (if we have exactly $d$ chunks). The original data $D$ can be reconstructed by inverting $G'$, which essentially implements a Lagrange interpolation using a matrix:

$$D = (G')^{-1} \cdot C' \tag{4}$$

Algorithm 1 presents the pseudocode of the encoding (line 1), decoding (line 15), and hashing (line 9) processes.

---

**Algorithm 1** Encoding, Decoding, and Hashing.

1: **function** ENCODE(*data*, *No.dataChunks*, *No.parityChunks*)
2:     *Chunks* ← *Split*(*data*)
3:     *chunkSize* ← length(*data*) / *No.dataChunks*
4:     *G* ← generator matrix(*x*, *No.dataChunks*, *No.parityChunks*)
5:     **for** *k* ← 0 to length(*chunks*) **do**
6:         *vector$_d$* ← extracting the data symbols from *chunk*(*k*)
7:         *Parity* ← *G* · *vector$_d$* calculating parity
8:         *encodeddata* ← concatenate(*vector$_d$*, *Parity*)
9:         *Hash* ← Calculated MD5 hash of *chunk*[*k*]
10:         *metadata.ChunkHashes*[*k*] ← *Hash*
11:         *metadata.ChunkOrders*[*i*] ← *k*
12:     **end for**
13:     **return** *encodeddata*, *metadata*
14: **end function**
15: **function** DECODE(*EncodedData*, *metadata*)
16:     *C′* ← *EncodedData*
17:     *d* = *metadata.datachunks*
18:     *p* = *metadata.paritychunks*
19:     *n* = *d* + *p*
20:     **for** *k* ← 0 to length(*EncodedData*) **do**
21:         *Hash* ← Calculate hash*chunk*[*k*]
22:         **if** *Hash* ≠ *metadata.ChunkHashes*[*k*] **then**
23:             locationCurruptedChunk[k] = metadata.ChunkOrder[k]
24:         **end if**
25:     **end for**
26:     **if** *length*(*locationCurruptedChunk*) ≠ 0 **then**
27:         $G'_{(d*d)}$ ← selecting any d rows out of n to form a square matrix
28:         $(G')^{-1}$ ← Calculating the inverse
29:         $D = (G')^{-1} \cdot C'$ ← Reconstructing the data matrix
30:     **end if**
31:     **return** *D*
32: **end function**
33: **function** GENERATORMATRIX(*x*, *rows*, *columns*)
34:     **for** *i* ← 0 to rows-1 **do**
35:         **for** *j* ← 0 to columns-1 **do**
36:             $G[i][j]$ ← $x_{(i \cdot j)}$
37:         **end for**
38:     **end for**
39:     **return** *G*
40: **end function**

---

### 3.3. Distributed Swarm Workflow Scheduler

After dividing the file into chunks, the next step is to select DSN (distributed storage network) service providers for storing these chunks. When a storage request is initiated, the system prefers to use the resources of the local region to which the user belongs (home region) until the local region has reached its capacity. In the proposed DSWS (distributed swarm workflow scheduler) approach, the system executes particle swarm optimization (PSO) in the local region as well as neighboring regions in parallel to find the best service providers in each of the regions.

PSO is an exploratory bio-inspired algorithm that helps to find the optimal solution from a large population of candidate solutions [32]. Each PSO particle indicates a potential solution to the problem of allocating file chunks to a storage node in a storage network. Here, we have a swarm of workflows (*d* + *p* chunks) and a pool of storage nodes. The goal is to discover the best storage node for each of the workflows. The structure of the

particles consists of a position, velocity, best-known position, and cost of that position. The particles indicate a potential solution for the file chunks across available storage nodes in the network. Each particle's position is initialized randomly because the randomness allows the algorithm to explore an extensive range of potential solutions during the iterations. The initial velocity of particles is set to zero, which indicates a neutral starting direction of movement for all particles in the search space. This is standard in PSO for allowing the algorithm to adapt the velocity dynamically depending upon both the particle's exposure and that of its neighbors. Since the objective is to minimize the cost function, the initial best cost is set to a very large value, indicating that any feasible solution will be better than this initial setting. The evaluation of a particle is performed based on the cost function (see Algorithm 2, lines 57–61). It is constructed to evaluate the suitability of storing file chunks on different IPFS nodes based on their available resources (CPU, storage space, and memory). The cost function ensures that nodes with higher available resources are preferred for providing the storage service.

Our DSWS (distributed swarm workflow scheduling) algorithm allows us to identify the regional and global best nodes from the entire network. The system employs the following methodology for the scheduling process, incorporating the particle swarm optimization (PSO) technique:

- Defining the fitness function: PSO requires a fitness function to assess solution quality. In the context of storing file chunks, this function is developed based on the cost of storing a specific chunk on a certain storage provider. The primary objective here is to find a service provider that minimizes the storage cost.
- Defining the search space: The search space consists of all potential solutions that the algorithm may explore. In this scenario, the search space contains all available storage providers in each of the regions that are capable of storing data chunks. It is important to periodically update the status and available resources of the storage providers for each execution. This adjustment is essential to indicate any modification in the search space.
- Executing the PSO in home and neighboring regions concurrently: Execute PSO to explore each search space concurrently to find regional and global best storage providers. It assesses the fitness function and identifies the storage providers that offer the most cost-efficient solution.

In general, in multidimensional search spaces, optimization algorithms such as ours do often not perform well because of the resulting computational complexity, specifically in a single-threaded environment. They might potentially also take a longer time to converge. Our proposed algorithm can evaluate multiple particles concurrently to scale well with the number of existing computational cores. In other words, DSWS can more effectively use available CPU resources by spreading the workload across multiple cores, thus reducing the time for convergence to an optimal solution.

Therefore, when there is a need to store some of the chunks in other regions, the incoming storage requests from the home region will be handled immediately. Moreover, DSWS also finds the global best service providers that are used for replication (Algorithm 2, line 19). The DSWS algorithm makes sure that data are stored as close to the user as possible by first locating the best storage nodes within local regions (Algorithm 2, line 8), reducing latency, and possibly lowering communication costs. Figures 3 and 4 explain the overall functionality of the Robust-DSN. Local optimization helps to balance the storage load across regions and prevents any single region from becoming a bottleneck, especially when a region does not have the required resources or becomes overburdened and there is a need to store some of the chunks in other regions. DSWS facilitates the distributed replication that enables the separation of the failure domain. Storing replicas at the globally best nodes, separate from the original data location, ensures that data are not only stored across different nodes but also across diverse geographic and network domains. This further reduces the risk of simultaneous failures affecting all data chunks. Algorithm 2 shows the pseudocode of the DSWS algorithm.

---

**Algorithm 2** Distributed Swarm Workflow Scheduler.

---

1: **function** DSWS(*wList*, *nodesInAllRegions*)
2:     *regionalBest* ← initialize empty list of NodeListWithCost
3:     **for** each *nodeGroup* in *nodesInAllRegions* **do**
4:         *nodeMat* ← PSO(*wList*, *nodeGroup*)
5:         **for** *p* in *nodeMat* **do**
6:             Sort *nodeMat*[*p*] based on cost
7:         **end for**
8:         *regBest* ← append*nodeMat* to *regBest*
9:     **end for**
10:     *homeRegionNodes*, *homeRegionIndex*, *found* ← initialize variables
11:     *listOfAllNodes* ← aggregate all nodes from *regionalBest*
12:     update *homeRegionNodes*, *homeRegionIndex*, *found*
13:     Sort *listOfAllNodes* based on cost
14:     *Map* ← map to track selected nodes
15:     Mark nodes in *homeRegionNodes* as selected in *Map*
16:     *globalBest* ← initialize empty list of NodeListWithCost
17:     **for** each *node* in *listOfAllNodes* **do**
18:         **if** *node* is not in *Map* **then**
19:             *globalBest* ← append *node* to *globalBest*
20:         **end if**
21:     **end for**
22:     **return** *regionalBest*, *globalBest*
23: **end function**
24: **function** PSO(*wList*, *nList*)
25:     *particles* ← **Particle structure** (position, velocity, bestPosition, bestCost)
26:     *particles* ← *population*(*PopSize*, *len*(*nList*), *len*(*wList*))
27:     **for each** *i* **in** range(*particles*) **do**
28:         particles[*i*].position ← rand.Float32()         ▷ Randomly initialized
29:         particles[*i*].velocity ← 0         ▷ Starting velocity
30:         particles[*i*].bestPosition ← 0         ▷ Will be updated during iterations
31:         particles[*i*].bestCost ← (>> 1)     ▷ Since we minimize the cost function, set possible maximum value
32:     **end for**
33:     *gBest* ← *arrayOfZeros*(*length*(*wList*))
34:     *gBestCost* ← *maxFloat64Value*
35:     *convergence* ← *newArrayOfSize*(*maxGenerations*)
36:     **for** *i* ← 0 **to** *maxGenerations* − 1 **do**
37:         **for** *j* ← 0 **to** *length*(*particles*) − 1 **do**
38:             **Concurrently do for particle** *j*:
39:             *p* ← *particles*[*j*]
40:             *p.Cost* ← *EVALUATEPARTICLE*(*wList*, *nList*)
41:             **if** *p.Cost* < *p.PBestCost* **then**
42:                 *p.PBestCost* ← *p.Cost*
43:                 *p.PBest* ← *copyOf*(*p.Position*)
44:             **end if**
45:             **if** *p.Cost* < *gBestCost* **then**
46:                 *gBestCost* ← *p.Cost*
47:                 *gBest* ← *copyOf*(*p.Position*)
48:             **end if**
49:         **end for**
50:         **for** *j* ← 0 **to** *length*(*particles*) − 1 **do**
51:             *updateVel* − *Pos*(*particles*[*j*], *gBest*, *length*(*nList*))
52:         **end for**
53:         *convergence*[*i*] ← *gBestCost*
54:     **end for**
55:     **return** *NodesBasedOnSuitability*(*gBest*, *wList*, *nList*)
56: **end function**
57: **function** EVALUATEPARTICLE(*w*, *n*)
58:     **if** *n.cpu* ≥ *w.cpu* & *n.mem* ≥ *w.mem* & *n.ram* ≥ *w.ram* **then**
59:         **return** *w.cpu*/*n.cpu* + *w.ram*/*n.ram* + *w.mem*/*n.mem*
60:     **end if**
61: **end function**

---

Communication with the DSN service providers is controlled through an API advertised by the DSN cluster, which organizes the functions of file uploading and downloading. This API plays an important role in accessing the current state of the service providers and their available resources and assists in the communication and management of processes within the DSN system. To participate in this ecosystem, any peer can operate as an access point for the DSN using an API to enable interaction. For enhanced resilience and to mitigate the risks of a single point of failure, multiple entry points can be configured, ensuring a continuous operation even if some of the entry points fail. By configuring multiple APIs and spreading entry points, we established a resilient and reliable distributed storage system capable of tolerating potential disruptions. This process can be seen in the `APIHANDLE` function in Algorithm 3, line 21.

---

**Algorithm 3** Distribute Chunks

---

1: **function** DISTRIBUTEDATA($encodedData, regionalBest, globalBest, metadata, api$)
2:     **for** each $i, data$ in $encodedData$ **do**
3:         **Concurrently do for data** $i$:
4:         $buffer \leftarrow$ new buffer with $data$
5:         $fileHash \leftarrow$ add buffer to IPFS using $api$
6:         **if** $fileHash$ generation failed **then**
7:             **Log error and exit**
8:         **end if**
9:         $rgBest \leftarrow regionalBest[i].peer$
10:        $gbBest \leftarrow globalBest[i].peer$
11:        $response \leftarrow$ apiHandle with $fileHash, rgBest, gbBest$
12:        **if** $response$ generation failed **then**
13:            **Log error and exit**
14:        **end if**
15:        $metadata.IpfsHashes[i] \leftarrow fileHash$
16:        **Print** "Sending data to storage node"
17:     **end for**
18:     **Wait for all data to be distributed**
19:     SaveMetadata($metadata$)
20: **end function**
21: **function** APIHANDLE($fileHash, rgBest, gbBest$)
22:     $addresses \leftarrow$ list of API addresses
23:     $lastError \leftarrow$ initialize as nil
24:     **for** each $address$ in $addresses$ **do**
25:         $url \leftarrow$ construct URL with $fileHash, rgBest, gbBest$
26:         $response, err \leftarrow$ POST request to $url$
27:         **if** $err$ is not nil **then**
28:             $lastError \leftarrow$ update error with $err$
29:             **continue to next address**
30:         **end if**
31:         **return** $response$, nil
32:     **end for**
33:     **return** nil, $lastError$
34: **end function**

---

### 3.4. Uploading Process

The data uploading process can be seen in Figure 3. To reduce the complexity of the flowchart, the process of chunk distribution to the regional best nodes is explained in a separate flowchart (Figure 4). In the uploading process, when a user submits a storage request, the system uses Reed–Solomon encoding (Algorithm 1, line 1) to split that data into multiple chunks (data and parity). A Reed–Solomon encoder takes a block of digital data and divides it into equal-sized data chunks. After this, by applying arithmetic operations, parity chunks are calculated. The parity chunks are extra redundant bits to deal with the chunk loss. An objective here is to tolerate the loss that could be caused by the distributed nature of the chosen platform and the resulting failure of some nodes. In that case, a fresh

copy of the status and the available resources of each solution space (home and neighboring regions) is fetched via the API so that the information can be used by DSWS. Here, the distributed swarm workflow scheduler DSWS (Algorithm 2) is executed, which uses the required resources of each of the chunks and scans all solution spaces concurrently to identify the regional and global best service providers. Once regional and global best DSN service providers are found, one copy of the chunks will be distributed to the regional best by giving preference to the home region, and replicas will be distributed to the global best (Algorithm 3, line 11). Algorithm 3 (line 3) concurrently executes the chunk distribution task using a lightweight multithreading technique and responds to the user for task completion. Each of the storage requests is processed asynchronously to parallelize the process of storing data chunks on the storage nodes based on available logical cores of the CPU, improving the system performance.

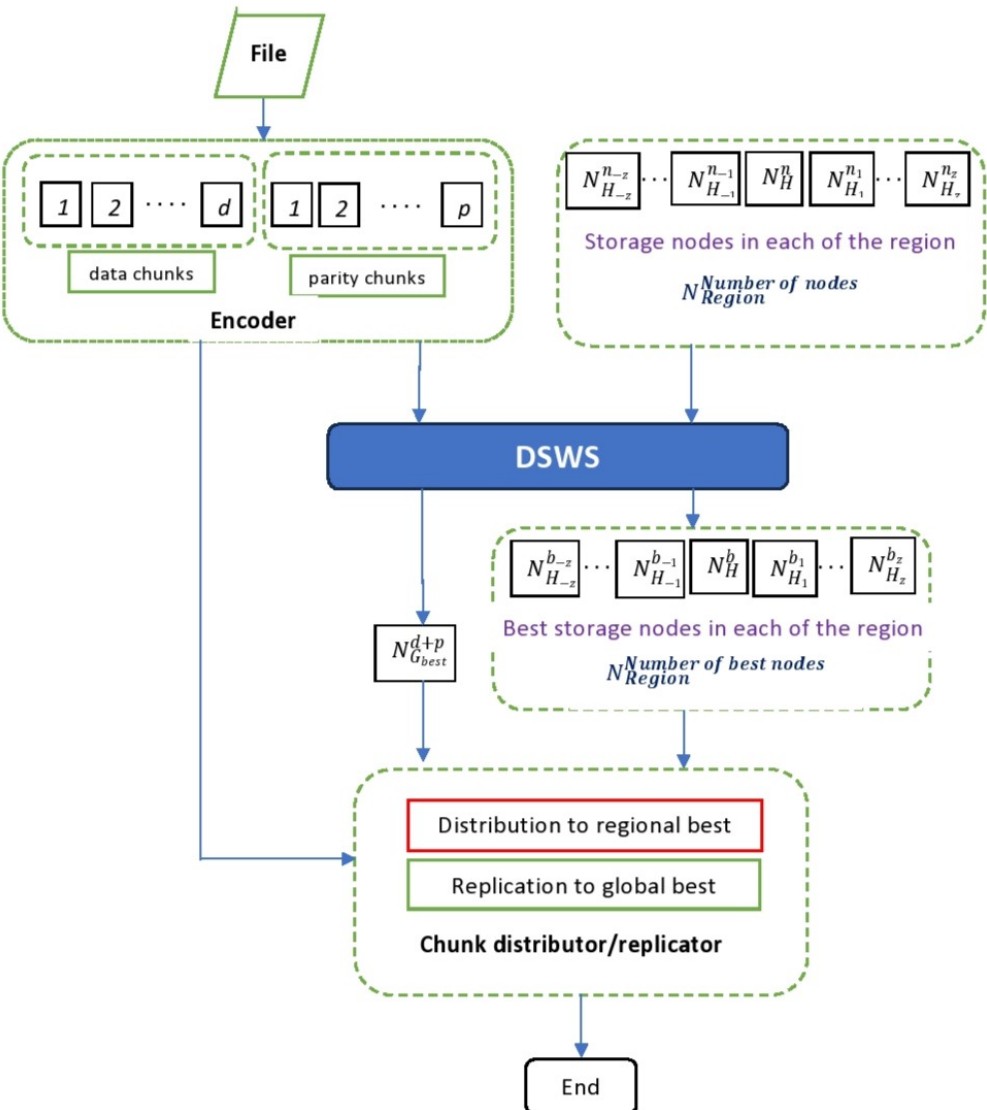

**Figure 3.** General storage workflow.

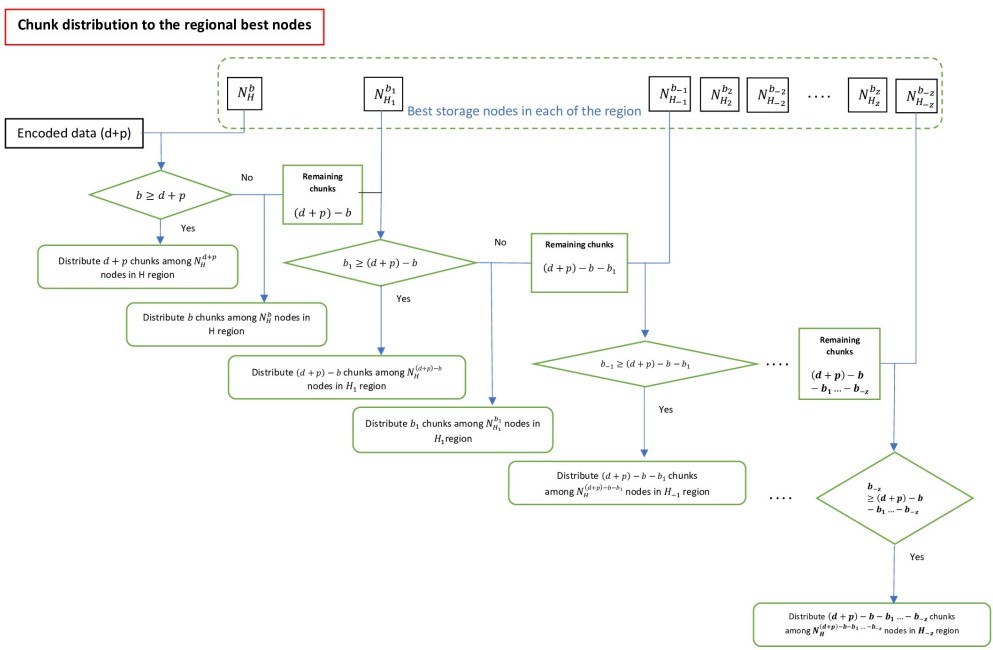

**Figure 4.** Detailed chunk distribution.

The Go programming language offers a powerful feature of goroutines, providing lightweight multithreading for running pieces of code concurrently. Goroutines are controlled by the Go runtime scheduler, which multiplexes routines onto smaller operating system threads and then executes on the available logical cores of the CPU. A logical core generally indicates a hardware thread. In processors with hyperthreading, each physical core can execute two threads, expanding the number of logical cores. However, whether each of the goroutines runs on independent CPU cores entirely depends on the number of available CPU cores. By default, the scheduler detects the number of available logical cores including those offered by hyperthreading. This enables the scheduler to fully utilize the availability of the CPU for concurrent processing. Usually, the operating system treads are 2 MB in size, while the the size of the goroutine is only 2 KB. Therefore, goroutines are very light compared with threads that enable us to handle hundreds of goroutines at the same time. Implementing the particle swarm optimization (PSO) algorithm and chunks distribution and fetching to and from IPFS storage nodes within our system are designed to run concurrently using these goroutines. Each PSO execution, which corresponds to determining an optimal node allocation for the given workflow, is encapsulated within a separate goroutine, allowing multiple PSO processes to execute in parallel depending upon the availability of logical cores. In our experiment, we used a Core i5-10210U CPU that contains eight logical cores because of hyperthreading (four physical cores). This lightweight multithreading approach reduces the time taken to find optimal solutions and maximizes the utilization of available computational resources.

*3.5. Downloading Process*

The download procedure involves two important functions: `getdata` and `downloadchunks` (Algorithm 4). The `getdata` function retrieves the IPFS hashes using the file name provided by the user from the metadata of the desired file (Algorithm 4, line 4). These hashes are then used by the API advertised by the storage network for asynchronously requesting the chunks, employing lightweight multithreading (line 9). The `downloadchunks` function fetches all chunks asynchronously and verifies the validity of each received chunk by computing its hash and comparing it with the associated metadata-stored hash. Algorithm 4 is designed to wait only for the minimum number of valid chunks required for file reconstruction (equivalent to the number of data chunks). As soon as this threshold of the minimum required number of valid chunks is met, ongoing

and remaining threads beyond this requirement are ended (line 27). Finally, Algorithm 4 applies the decoding process to reconstruct the original data (line 16).

---

**Algorithm 4** Download Data

---

1: **function** GETDATA
2:     **Print** "Provide the name of the file you want to retrieve:"
3:     *fileName* ← read user input
4:     *metadata* ← LoadMetadata(*fileName*)
5:     *api* ← initialize new shell with localhost:5001
6:     Create context *ctx* and cancel function *cancel*
7:     *done* ← create channel for signaling completion
8:     Start a new concurrent process:
9:         *Chunks*, *No.ValidChunks* ← DOWNLOADCHUNKS(*ctx*, *cancel*, *metadata*, *api*, *done*)
10:     **if** *No.validChunks* < *metadata.DataShards* **then**
11:         **Print** "Not enough valid chunks to reconstruct the file"
12:         **Exit**
13:     **end if**
14:     Sort *results* based on index
15:     Copy data from *results* to *retrievedData*
16:     *Data* ← decodingData(*Chunks*, *metadata*)
17:     **Comment** "Save the decoded Data"
18: **end function**
19: **function** DOWNLOADCHUNKS(*ctx*, *cancel*, *metadata*, *api*, *done*)
20:     *results* ← initialize empty array for chunk results
21:     *validChunks* ← 0
22:     **for** each *i* in range of *metadata.IpfsHashes* **do**
23:         Start a new concurrent process for index *i*:
24:         Retrieve data from IPFS using *metadata.IpfsHashes*[*i*]
25:         Validate retrieved data
26:         Update *results* and *No.validChunks* accordingly
27:         If enough valid chunks are retrieved, signal completion
28:     **end for**
29:     **return** *results*, *No.validChunks*
30: **end function**

---

## 4. Evaluation

To demonstrate the effectiveness of our proposed system, we carried out a comprehensive evaluation of our Robust-DSN solution. Our system provides the users with the flexibility to adjust the parameters to enhance both redundancy and performance according to explicit requirements. We benchmarked our Robust-DSN with full uniform replication, where every chunk has the same number of copies. Our assessment includes individual and joint evaluations of HYDREN and DSWS.

Our first evaluation focuses on the probability of data loss in Section 4.1. The next evaluation in Section 4.2 addresses assessing the effectiveness of HYDREN in ensuring data availability and encoding time by benchmarking with a state-of-the-art solution. Following this, in Section 4.4, we evaluate our optimization algorithm, DSWS (distributed swarm workflow scheduler), in terms of convergence time. Finally, in Section 4.5, we evaluate our system for overall data uploading and downloading performance. The fault tolerance of our algorithm is evaluated by examining the probability of data availability in different failure scenarios.

### 4.1. Evaluation of Probability of Data Loss

To formulate the probability of data loss with a hybrid distributed replication and encoding network, we need to define the following parameters:

- $N$ ← The total number of storage nodes in the system.
- $n$ ← The number of nodes storing file chunks.
- $P_{\text{fail}}$ ← The probability of a node failing.

- $P_{\text{available}} = 1 - P_{\text{fail}}$ is the probability of a node being available or operational.

Our objective is to minimize the probability of data loss, since without using any data persistence approach to deal with fault tolerance, all chunks are required for data recovery. Thus, the probability of preserving all chunks is the product of the individual probabilities:

$$P_{\text{all chunks available}} = (1 - P_{\text{fail}})^n \tag{5}$$

Therefore, we can say that the probability of missing at least one chunk can be calculated as follows:

$$P_{\text{missing at least one chunk}} = 1 - (1 - P_{\text{fail}})^n = 1 - P_{\text{available}}^n \tag{6}$$

It is worth noticing that as $n$ increases, $P_{\text{available}}$ decreases quickly. If we need more nodes to reconstruct a file, the probability of all of them being available decreases exponentially. This situation can be mitigated by implementing data persistence algorithms such as encoding and replication mechanisms to improve overall file availability.

4.1.1. Probability of Data Loss Using Rs-Encoding

For Reed–Solomon encoding, the data are split into $d$ chunks and calculated, and $p$ parity chunks are added, making a total of $d + p$ chunks. Only $d$ out of $d + p$ chunks are needed to reconstruct the original data.

In this scenario, the objective is to compute the probability of losing enough chunks to make the data reconstruction impossible. Since we need at least $d$ out of $d + p$ chunks, we can say that if we lose more than $p$ chunks, the file cannot be reconstructed. The range can be $p + 1, p + 2, \ldots, p + d$ chunks.

$P_{\text{fail}}$ the probability of losing a single chunk, which corresponds to the probability of the failure of the node which is storing that chunk. So, the probability of losing any specific set of $k$ chunks can be written as $P_{\text{fail}}^k$. If there are a total of d+p chunks, the number of ways to choose $k$ can be expressed as a combination formula $\binom{d+p}{k}$. The total probability of losing more than p chunks is the sum of probabilities for each $k$ that ranges from $p + 1, p + 2, \ldots, p + d$. Thus, the probability of losing enough chunks such that the data reconstruction is impossible can be expressed as

$$P_{\text{data loss}}(Encoding) = \sum_{k=p+1}^{d+p} \binom{d+p}{k} P_{\text{fail}}^k \tag{7}$$

Figure 5 illustrates, by keeping node failure probability set to 30, how the probability of data loss changes by varying the number of data and parity chunks in a distributed storage system using Reed–Solomon encoding only. From the color slope, which denotes the likelihood of data loss, we can see that the probability of data loss reduces as the number of parity chunks grows. This was expected since additional parity chunks indicate more redundancy and, consequently, a greater tolerance for node failure. Conversely, for a constant number of parity chunks, the probability of data loss also rises as the number of data chunks grows. This is because more data chunks could potentially be lost. Overall, the figure shows the trade-off between fault tolerance and storage efficiency. By using more parity chunks, the system shows more robustness against data loss but at the cost of large storage overhead. Thus, the optimal balance between $p$ and $d$ may depend on the particular requirements of storage capacity and fault tolerance.

Interpreting storage usage, the total storage consumption $S$ for the data, applying Reed–Solomon encoding, can be calculated as

$$S = (p + d) \times c \tag{8}$$

where $c$ is the chunk size, $d$ are the data chunks, and $p$ represents the parity chunks. Although this method improves file availability to a certain extent, we cannot manage the loss of more than $p$ chunks.

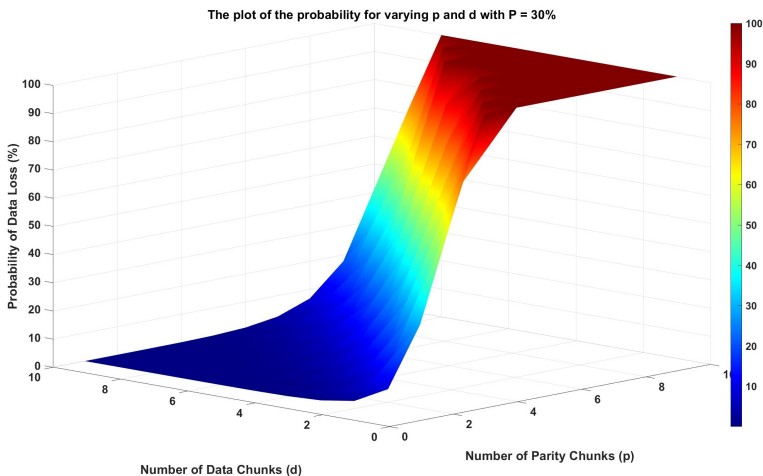

**Figure 5.** The probability of data loss for varying d and p.

### 4.1.2. Probability of Data Loss Using Replication

In case of replication, each data chunk is copied $x$ times to improve data availability. We have a total of d chunks, and each chunk is replicated $x$ times to make a total of $d \times (x)$ chunks. So, the probability of losing all the replicas of a single chunk can be expressed as $P_{\text{fail}}{}^{x}$. Since any of the d chunks could be the one that loses all its replicas, the total probability can be expressed as

$$P_{\text{data loss}}(Replication) = d \times (P_{\text{fail}})^{x} \tag{9}$$

Figure 6 shows, by keeping the node failure probability set to 30, how the probability of data loss changes by varying the replication factor and the number of data chunks in a distributed storage system using replication only. As the replication factor rises, the probability of data loss reduces, which is aligned with our expectation because more replicas of each data chunk decrease the probability of data loss. On the other hand, the probability of data loss increases as the quantity of data chunks increases because more chunks might potentially fail. Moreover, it also emphasizes the deteriorating effect of the replication factor, as the probability of data loss rapidly increases exclusively for lower replication factors.

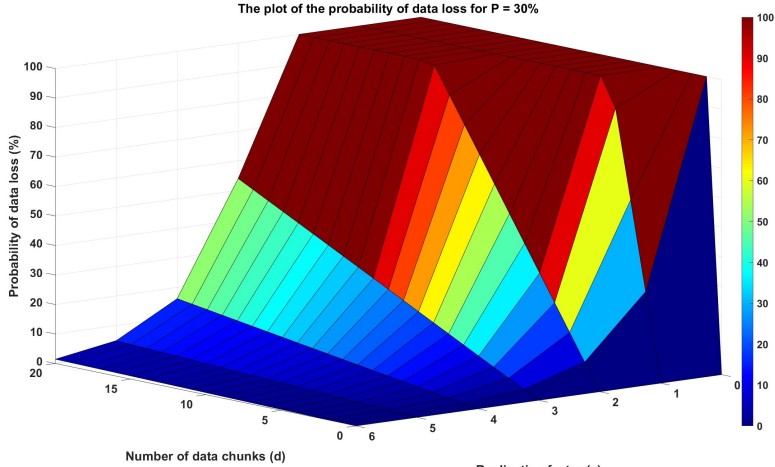

**Figure 6.** The probability of data loss vs. d and x.

Interpreting storage usage, the total storage consumption $S$ for the data, applying replication, can be calculated as

$$S = c \times d \times x \tag{10}$$

where $S$ is the total storage utilization of a file when we use replication, $c$ is the chunk size, $d$ represents the data chunks, and $x$ represents the replication factor.

### 4.1.3. Probability of Data Loss Using Hydren

If we use only replication, the file becomes irrecoverable if all $(x)$ replicas of even a single chunk are lost. On the other hand, for the encoding method, the file will be irrecoverable in the case of losing more than $p$ chunks. To deal with a large-scale failure (many nodes going down) with minimum storage overhead, a hybrid approach can be tuned to provide a balance between storage efficiency and failure probability. In this approach, a file is split into $d$ chunks and calculated, and $p$ parity chunks are added, making a total of $d + p$ chunks. After that, each of the chunks is replicated $x$ times to make a total of $x(d + p)$ chunks of a file. For a file to be lost in this hybrid approach, we must lose all the $(x)$ replicas of more than $p$ chunks. As described in Section 3.5 (replication only), the probability of losing all the replicas of a single chunk can be expressed as $(P_{\text{fail}})^x$. The probability of losing all replicas of $k$ chunks can be expressed as $((P_{\text{fail}})^x)^k$. Moreover, we have a total of $d + p$ chunks; the number of ways to choose $k$ can be expressed as a combination formula $\binom{d+p}{k}$. So, the total probability of losing more than $p$ chunks with their replicas at the same time can be expressed as

$$P_{\text{data loss}}(HYDREN) = \sum_{k=p+1}^{d+p} \binom{d+p}{k} (P_{\text{fail}}^x)^k \tag{11}$$

Interpreting storage usage, the total storage consumption $S$ for the data, applying Reed–Solomon encoding, can be calculated as follows:

$$S = (p + d) \times c \times x \tag{12}$$

where $c$ is the chunk size, $x$ is the replication factor, $d$ represents the data chunks, and $p$ represents the parity chunks. Although this method improves file availability to a certain extent, we cannot afford the loss of more than $p$ chunks.

Figures 7 and 8 demonstrate the probability of data loss using HYDREN, which employs distributed replication and encoding. The graphs highlight the significance of parity chunks in protecting the data. There is an optimal bound for parity chunks that can minimize the probability of data loss efficiently, keeping the replication factor constant. From Figure 7, which uses replication factor 2 only, it can be seen that even with a parity-to-data chunk ratio of 1:6, the system provides a probability of data loss of 0.2, which shows significant improvement in data availability and effectiveness of the HYDREN approach. Moreover, Figure 8 describes the impact on the probability of data loss when we increase the replication factor to 3. Figure 8 shows that even with a parity-to-data chunk ratio of 1:10, the probability of data loss is around 0.04. With the increase in replication factor, each chunk (data and parity) has more replicas and is distributed across the regions, which have different failure rates and should in principle facilitate the system in dealing with large-scale failures.

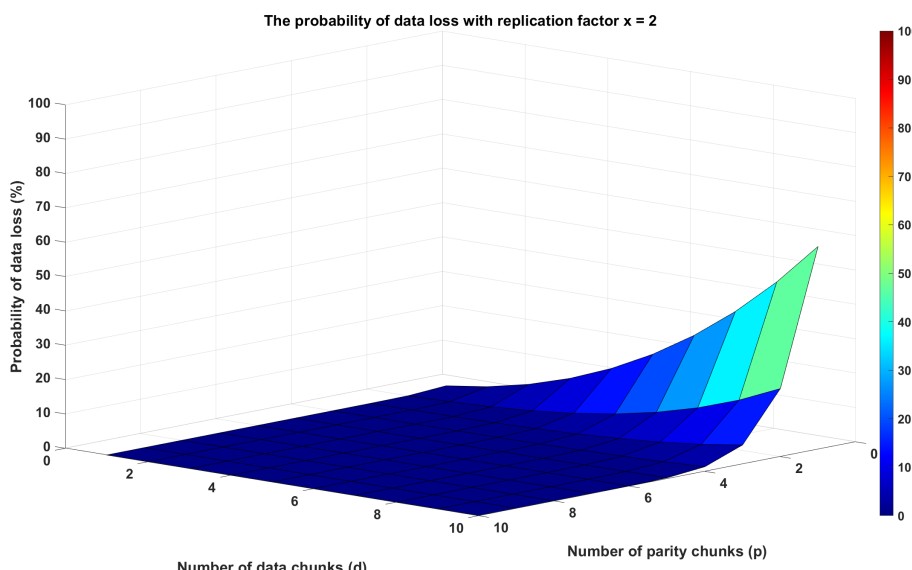

**Figure 7.** The probability of data loss, varying d and p with replication factor 2.

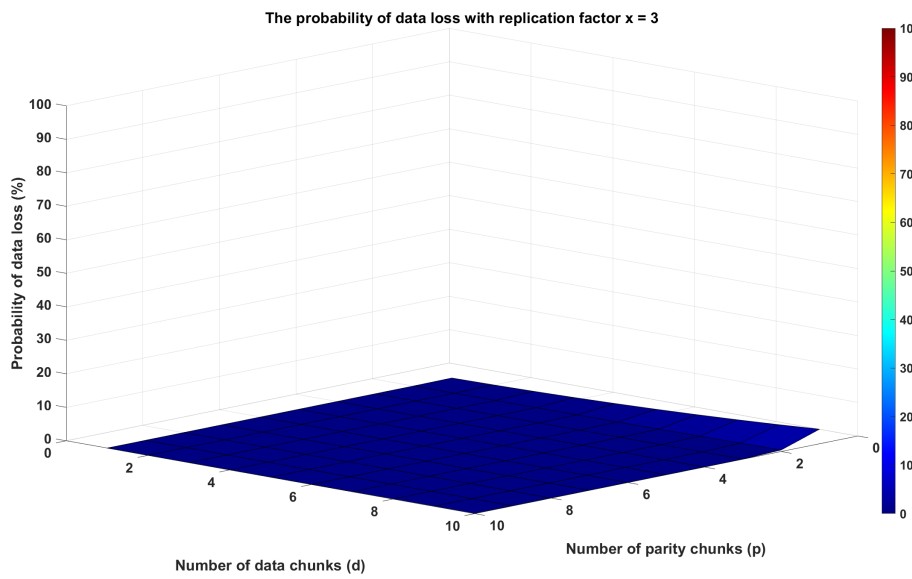

**Figure 8.** The probability of data loss, varying d and p with replication factor 3.

*4.2. Evaluation of Hydren Data Availability and Encoding Speed*

We evaluated HYDREN for data availability and encoding speed by benchmarking it with existing systems.

Data Availability

Figure 9 demonstrates the data recovery likelihood versus the peer failure rate for three distinct storage systems: HYDREN, Swarm network, and Snarl [33]. The Swarm network is a decentralized, self-sustaining, undelaying infrastructure of storage for the Ethereum ecosystem. The Swarm network allows the storage and distribution of data across a peer-to-peer network of nodes [34]. Snarl is an overlay network that creates a logical network on top of Swarm and acts as an additional layer between the user and the storage network. Figure 9 illustrates that the data recovery likelihood for all the systems reduces as the peer failure rate increases. For this experiment, a 100 MB file was taken to evaluate the systems. Thus, for 100 MB of data, HYDREN utilized 506.66 MB for configuration ($r = 4$, $d = 15$, and $p = 4$) and 1405.55 MB for configuration ($r = 11$, $d = 18$, and $p = 5$). The

blue line plot, indicating HYDREN, sustains a data recovery likelihood close to 100% even if the peer failure rate grows to around 50% for configuration ($r = 4$, $d = 15$, and $p = 4$) and 79% for configuration ($r = 11$, $d = 18$, and $p = 5$). This indicates that HYDREN is highly robust to peer failure and can maintain excellent data availability even in large-scale peer failure circumstances.

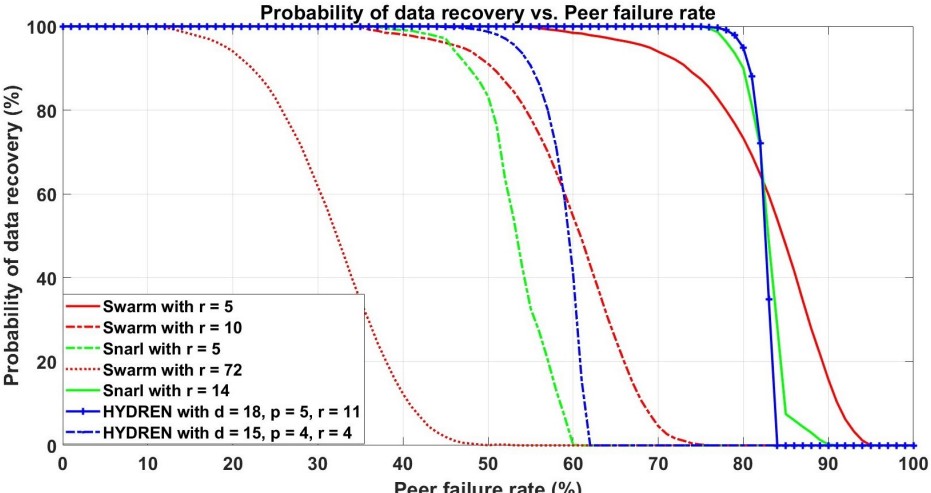

**Figure 9.** File recovery likelihood vs. peer failure rate.

Conversely, the green and red plots represent Snarl and Swarm, respectively, indicating a quick deterioration in the data recovery likelihood as the peer failure rate grows. Our proposed system shows dual advantages against the Swarm network. For a peer failure rate of 50%, saving around half of the storage, the proposed system provides about 8% higher data recovery likelihood compared with the Swarm network for replication factor 10. Likewise, for a peer failure rate of 80%, the proposed system uses 5.122 times less storage and provides about 21.6% higher data recovery likelihood compared with the Swarm network with $r = 72$. Furthermore, Snarl also cannot match the effectiveness of HYDREN here. HYDREN and Snarl consume around the same storage. For a peer failure rate of 50%, the data recovery likelihood of the proposed system is about 100%, while Snarl provides 83.3%. Similarly, for a peer failure rate of 80%, the data recovery likelihood of the proposed system is about 95%, while Snarl gives 90.6%.

### 4.3. Data Integrity

The uploaded file is encoded into multiple pieces when a storage call is initiated. The system calculates the hashes of all chunks that are used as unique fingerprints to validate the integrity of each chunk. In the downloading procedure, as a chunk is fetched from the storage node, it is verified by recomputing and comparing its hash with the associated reference hash. If a retrieved chunk is identified as corrupted or tampered with, it is leftover, and the system carries over to the next chunk. This procedure makes sure that only validated chunks are considered to be a part of the file reconstruction process. To improve the system's performance and minimize the latency, the algorithm stops the process of integrity verification as soon as it collects an adequate number of valid chunks; specifically, this is the number equivalent to the data chunks, which is the minimum required number of chunks for file reconstruction. This facilitates resource optimization and fast processing, as unessential verification of chunks is prevented once the required proportion of valid chunks is obtained.

### Encoding Speed

We also evaluated the effectiveness of HYDREN by assessing its encoding speed and comparing it against Snarl as a state-of-the-art solution. Figure 10 provides a 3D vision

of HYDREN's encoding speeds using a 10 MB file, modifying both the parity and data chunks. For the smaller number of parity chunks, the encoding time stays reasonable across a spectrum of data chunks, showing only an insignificant increase. However, there is a substantial increase in encoding time with a rise in parity chunks, which was expected because of the additional computation needed for generating the parity chunks. Table 2 shows a processing comparison between HYDREN and Snarl for different file sizes. Compared to Snarl, HYDREN consistently shows lower processing times for all file sizes, verifying its effectiveness. For a 1 MB file, HYDREN takes only 0.001854 s to encode, compared with Snarl's 0.0038 s. This tendency continues as file size increases. HYDREN takes only 0.017804 s to encode a 10 MB file, while Snarl encodes it in 0.037 s. The difference in encoding time reaches its peak for the 1000 MB file, where HYDREN encodes it in 1.719 s, better than Snarl's 3.600 s. Moreover, the benefit of HYDREN is considerable for larger data (Figure 11), where the encoding time for HYDREN does not grow larger proportionally with the size of the data, as opposed to Snarl.

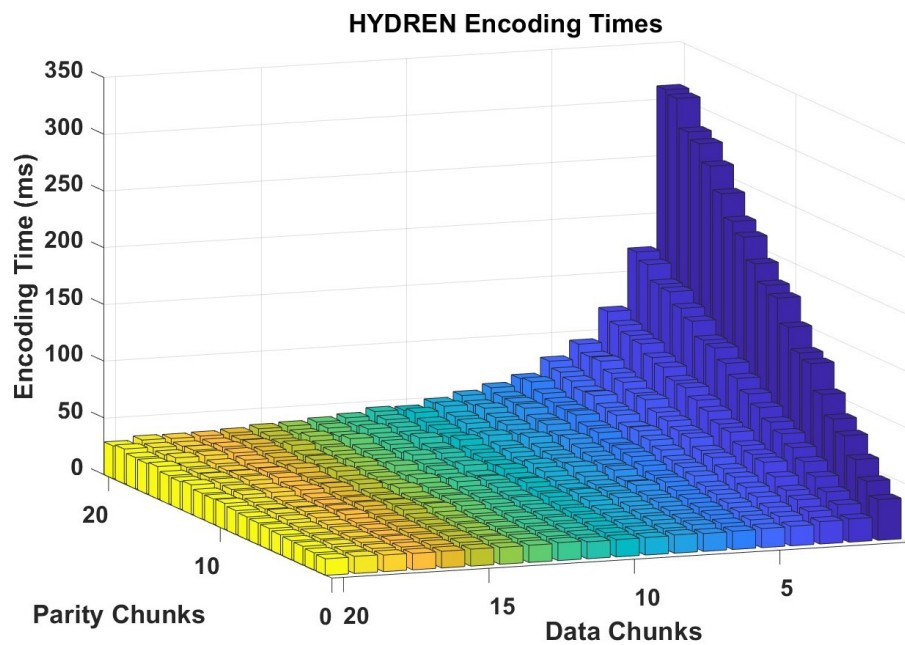

**Figure 10.** Encoding time of HYDREN for varying d and p using a 10 MB file.

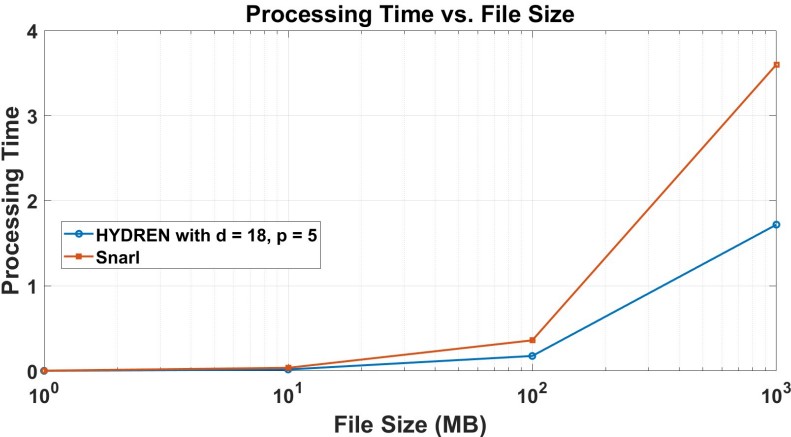

**Figure 11.** Encoding time comparison.

**Table 2.** Processing times comparison between Snarl and HYDREN.

| File Size (MB) | HYDREN Time (s) | Snarl Time (s) |
|:---:|:---:|:---:|
| 1 | 0.001854 | 0.0038 |
| 10 | 0.017804 | 0.037 |
| 100 | 0.176377 | 0.360 |
| 1000 | 1.719 | 3.600 |

The efficiency of HYDREN to maintain data availability can be seen in Figure 9. Its performance can be recognized by its robust encoding method, strategical replication approach, and effective node selection method, which incorporates regional and global optimization. This guarantees that even with a large-scale peer failure, there is adequate redundancy and mechanisms to recover the original data. This robustness shows HYDREN as a reliable distributed storage system appropriate for circumstances where data availability is important and peer failure can happen frequently or unpredictably. HYDREN also shows excellent scalability, which is a fundamental feature of distributed storage systems for handling ever-larger datasets. Thus, HYDREN is a robust and effective solution for improving data availability in distributed storage systems.

### 4.4. Evaluation of Dsws Performance

Figure 12 illustrates the performance of the DSWS distributed swarm workflow scheduler. The plot demonstrates the execution time of DSWS for varying numbers of service providers and workflows. It can be noted that the execution time increases linearly for both axes with the increase in the number of nodes and workflows, which was expected, since with the increase in workflows, more computational tasks are introduced and the search space for optimization algorithms grows larger.

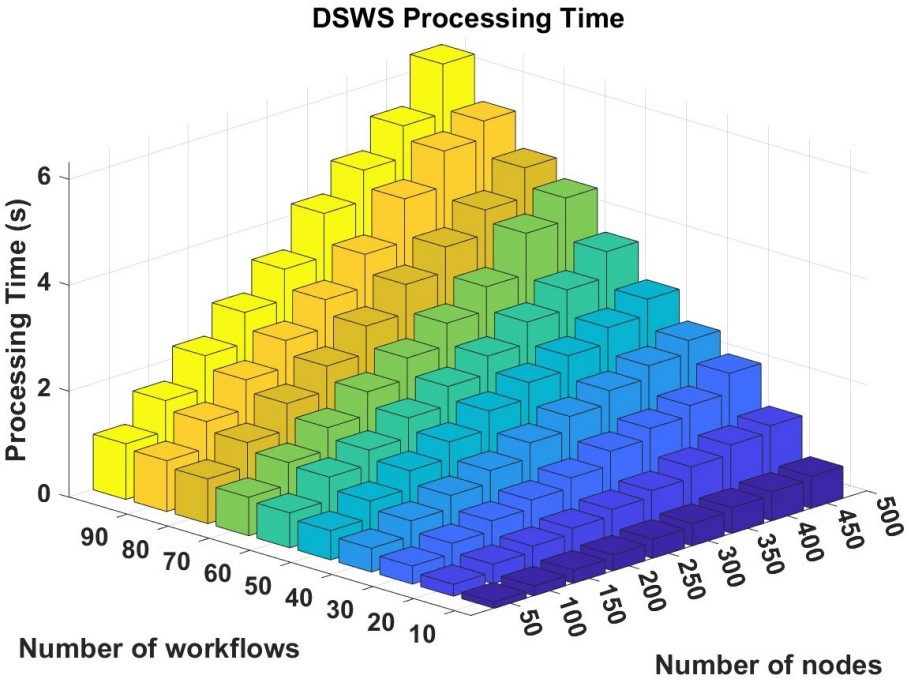

**Figure 12.** DSWS processing time.

Figure 13 compares the performance of DSWS with a genetic algorithm (GA) solution [35], particle swarm optimization (PSO) [36], and ant colony optimization (ACO) [37] for 30 workflows. The results indicate that DSWS continually outperforms the traditional algorithms, sustaining lower processing times for different sets of nodes. This suggests that

DSWS is efficient not only for distributing workflows but also for executing them more quickly than traditional methods.

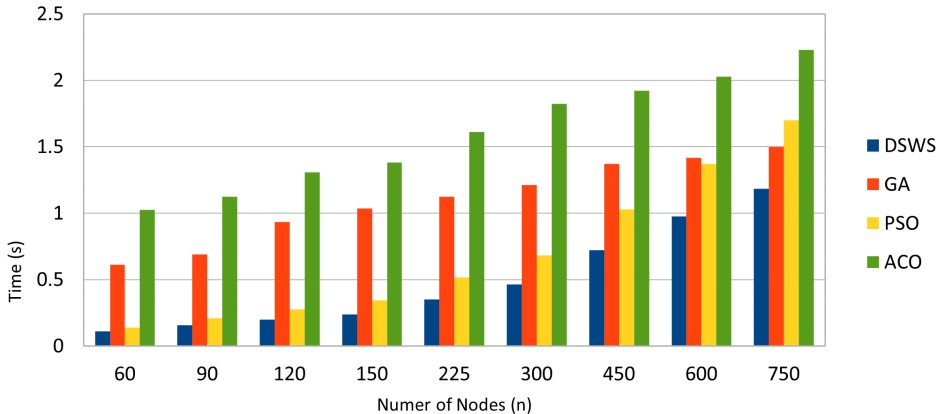

**Figure 13.** Processing time comparison for 30 workflows.

Figure 14 provides a convergence analogy, indicating how promptly each method proceeds toward an optimum solution. Compared with the PSO, GA, and ACO methods, the proposed DSWS algorithm converges faster and reaches the lowest cost function value in the least elapsed time, which is an illustration of a more effective optimization approach, arriving at an optimal or near-optimal solution in less time. Regardless of the size of the solution space, the proposed DSWS system not only improves processing times but also converges quickly to an optimal solution. This confirms DSWS as a significant improvement in distributed scheduling approaches.

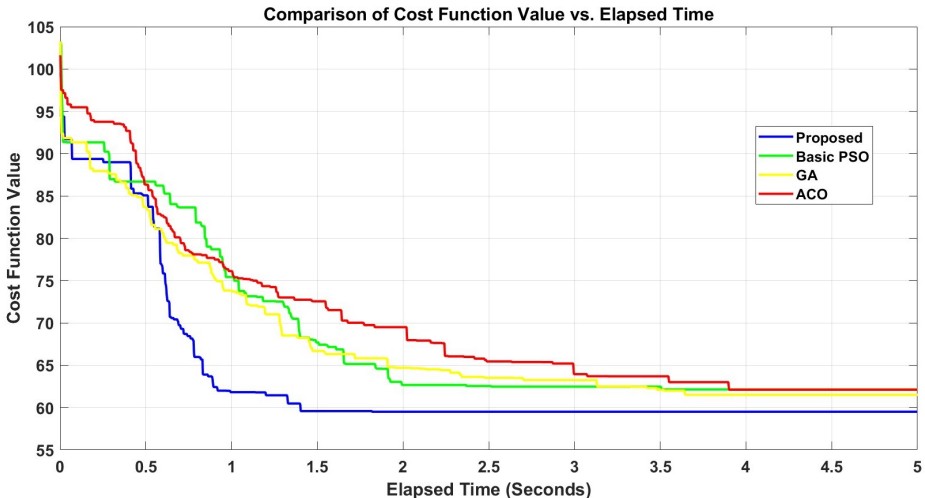

**Figure 14.** Convergence time comparison.

### *4.5. Evaluation of Robust-Dsn Execution Time*

To validate the capabilities of the proposed system, we did undertake a performance assessment by benchmarking against the baseline IPFS network. In this experiment, we built a distributed storage network using IPFS, deploying 20 independent Docker containers on a virtual machine with specifications of 16 GB RAM, 256 GB of storage, and a five-core CPU. All the nodes in the network were configured with distinct resources to mimic a network of heterogeneous service providers that use content-based addressing. Users could interact with the storage network through an API advertised by the network and deliver their data along with corresponding performance requirements to call on-demand and reliable storage services. This assessment was carried out for two operations: file uploading

and downloading. The performance indicator was the average time for uploading and downloading files of sizes varying from 10 MB to 1000 MB. In the case of IPFS, the file was uploaded with no replication. In comparison, the proposed system employed the HYDREN approach with the following parameters: ($d$ = 15, $p$ = 4). Each experiment was executed 15 times, and the average for both the uploading and downloading processes was determined.

Figures 15 and 16 indicate that for each of the experiments, the proposed Robust-DSN outperformed the IPFS in terms of file uploading and downloading times. The processing duration for both uploading and downloading files was also influenced by the parameters of encoding $d$ and $p$. As Figure 10 shows, for a fixed number of parity chunks, the encoding time reduced with an increase in the number of data chunks. However, we employed the following parameters: ($d$ = 15, $p$ = 4).

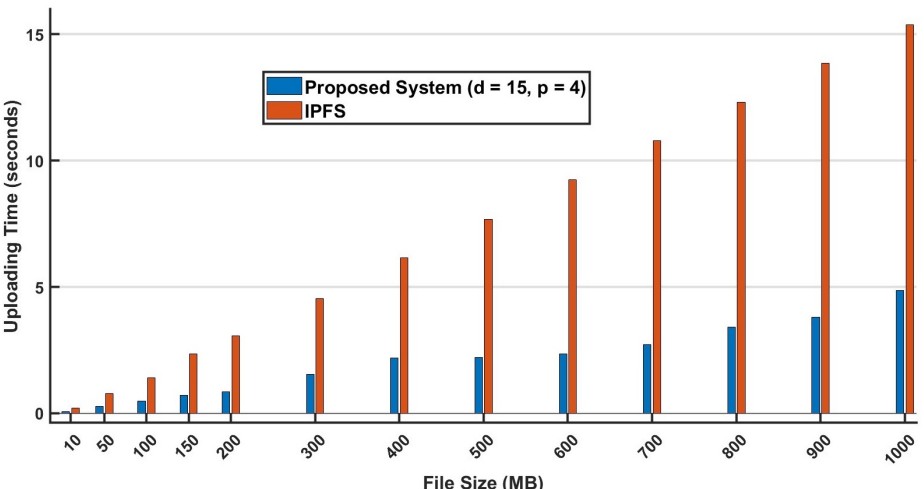

**Figure 15.** Uploading time comparison.

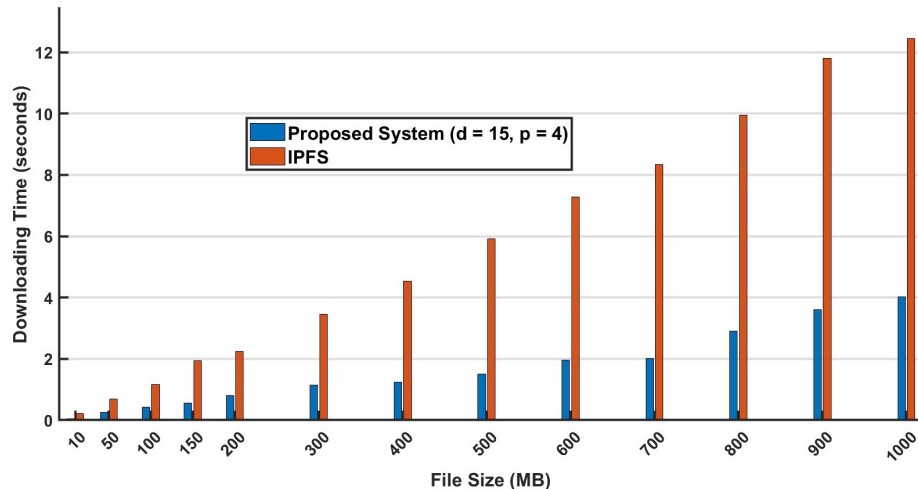

**Figure 16.** Downloading time comparison.

The results in Figure 15 show that both systems demonstrate that as the file size increases, there is also an increase in uploading time. However, the proposed system consistently maintains a lower uploading time compared to IPFS. This is because of the effectiveness of the chunk distribution approach enabled by a lightweight multithreading strategy that accelerates task completion and scales well even as the size of the file approaches 1000 MB. Conversely, the IPFS network executes this process with a single stream.

The file downloading time comparison can be seen in Figure 16, which shows the improved performance of Robust-DSN compared to IPFS for all file sizes. The IPFS system

retrieves the file via a single stream, whereas the proposed system fetches the file chunks asynchronously, employing lightweight multithreading. Moreover, the proposed system does not need a comprehensive set of uploaded chunks. Instead, it relies on an adequate subset of valid chunks (*d* number of chunks). The system waits only for d valid chunks and then terminates the remaining threads, thereby reducing both downloading and processing times.

Furthermore, our system provides an advanced mechanism for file distribution, while IPFS uploads files to its local daemon and propagates only the CID (content identifier) of the file using the DHT (distributed hash table) instead of transferring the actual file to other nodes for storage. In the Robust-DSN case, the file is split into data chunks and encoded to calculate parity chunks, suitable nodes are identified, and these chunks are actively distributed and replicated across nodes in different regions, which realizes the true sense of a distributed system. In IPFS, a single node hosts and advertises a complete file on the network. The intensity of the file distribution depends on the popularity of the CID in the network. A file can only be served by additional nodes if it has been accessed by these nodes via CID, and each node retains a full copy. Therefore, if a file is hosted by a single node and is unpopular, it would be inaccessible if the node failed.

## 5. Evaluation of System Constraints and Security Risks

For the architecture of the proposed DSN, we made some critical assumptions about the network contributors. we assume that all the nodes in the network are trustworthy and consistently reveal correct information about their available resources. Additionally, it is assumed that each of the service providers in the network holds equal privilege and can join either as a service provider or a user. This approach ensures a consistent policy for joining the network and improving the system's autonomous and decentralized nature. These assumptions are crucial to our system's operation, as accurate information on available resources is processed for data distribution and allocation. Though we employed an integrity-inspection process, where the hashes of all chunks are calculated before distribution and then afterward, at the downloading of the chunks, are recalculated and compared with the associated hash to validate the integrity of all the chunks, there is a lack of comprehensive security features. Although data are split into chunks and distributed to distinct nodes, the security data may be compromised if malicious nodes join the system. This is because we store the chunks in plaintext, exposing them to possible security breaches. Moreover, another security concern may appear specifically from the centralized management of metadata produced by the encoding scheme used in the system. It might turn into a single point of failure and become a potential target for attackers, compromising data availability. To alleviate this issue, there is a need for a robust authentication mechanism for participants and advanced encryption for the security of data chunks.

## 6. Conclusions

We presented a comprehensive solution for distributed storage systems to mitigate the issues of data availability, integrity, and task management. The overall Robust-DSN system incorporates HYDREN, a system developed for improving file availability through fast and reliable distributed replication and encoding, and DSWS, an advanced swarm intelligence-based workflow scheduler, as the main components, resulting in a robust DSN framework for distributed storage platforms.

The proposed architecture is established based on a layered approach, where the individual layer is responsible for a particular functionality. The data storage layer is based on the IPFS network. At the layer for handling data availability and integrity, HYDREN provides encoding and replicating of the data across home and neighboring regions for distributing the failure cause, certifying that data are not only protected but also readily accessible when required. This is specifically critical where data loss or unavailability can have significant repercussions. In addition to this, the DSWS layer provides a workflow scheduling approach, which helps to optimize the allocation of workflows across the

regional distribution of service providers. The experimental results provide evidence of DSWS's effectiveness since it outperformed traditional approaches like PSO, GA, and ACO in processing time and convergence rate. This is specifically significant in environments with a larger number of workflows, where DSWS sustains lower processing times across different sets of nodes. Moreover, the convergence plot confirms a faster convergence rate to optimal or near-optimal solutions. This is crucial for real-time applications where instant decision making is critical.

To conclude, Robust-DSN presents the following advancements in the design of distributed storage systems:

- Our system ensures data availability, even in case of network issues, hardware failures, or other disruptions. This is achieved by applying the HYDREN technique, where data are distributed in regional best nodes and replicated across global best nodes. This resiliency against large-scale failures leads to its robustness.
- The proposed system ensures the integrity of stored data by hashing each of the chunks and verifying them while downloading. This process prevents data corruption and ensures that data are not tampered with.
- The system can manage resources, schedule storage requests, and maintain performance under varying load conditions. This involves intelligent load balancing across regions and proactive workflow scheduling.
- The system can execute storage and retrieval requests instantly, reducing latency and processing times. This is due to an efficient data placement approach (lightweight multithreading) that concurrently performs the tasks.

The results show that compared with a state-of-the-art system, the proposed system offers 15% more file recovery likelihood, even for a peer failure rate of 50%. Moreover, with the configuration of replication factor 4 and the same failure resilience as IPFS, it saves 50% storage and offers 8% more file recovery likelihood. The cooperation between DSWS and HYDREN produces a balanced methodology for robustness, concentrating on both the static and dynamic characteristics of distributed storage systems. Future research directions include some important extensions and improvements to our present system. Firstly, our goal is to integrate blockchain technology into our existing framework to explore its ability for decentralized metadata management. Secondly, we would like to explore and implement encryption schemes for security and access management, thus ensuring a more robust and secure distributed storage system. Thirdly, we intend to incorporate an incentivization mechanism to motivate contributors in the storage network ecosystem. Moreover, this study is set to expand by acting as a storage platform for emerging technologies such as edge computing and the Internet of Things (IoT). The integration of these advanced technologies is likely to substantially improve the system's capabilities, thus expanding the proposed system's abilities in a wide range of applications.

**Author Contributions:** Z.H. devised the study idea and research questions, designed the methodology and experimental framework, collected and analyzed data and drafted the initial manuscript. H.R.B. assisted in analysis, assisted in developing the experimental framework, provided feedback during manuscript drafting, reviewed and edited the manuscript. N.E.I. helped to formulate and conceptualize the study, provided expertise in theoretical frameworks, assisted in results validation, participated in revision. C.P. supervised the overall research, provided support in designing methodology, provided valued input during discussions and improvements. All authors have read and agreed to the published version of the manuscript.

**Funding:** This research work is a part of the Ph.D. program at the Free University of Bolzano. There was no external funding received for this research or preparing this article. This research was conducted with the facilities and resources afforded by the Free University of Bolzano, with no individual grants from funding organizations or commercial or nonprofit sectors.

**Data Availability Statement:** The results and assessments presented in this article are based exclusively on the data produced during the research and do not use externally accessible datasets. The source code of the developed system in this research work can be provided upon request by the corresponding author.

**Conflicts of Interest:** The authors state that there are no competing interests or associations that could influence the work presented in this article. This research was conducted as a part of the Ph.D. program at the Free University of Bolzano without any financial relationships that could be a possible conflict of interest.

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
