# Peer review of "Robust-DSN: A Hybrid Distributed Replication and Encoding Network Grouped with a Distributed Swarm Workflow Scheduler"

_electronics, doi:10.3390/electronics13101861_

Round 1
Reviewer 1 Report
Comments and Suggestions for Authors
- In Section 3.2, provide more technical details and pseudocode for the encoding/decoding process beyond just the matrix operations.
- In Section 3.3 (DSWS Algorithm 2), clarify how the initialization of particles is done and what the evaluate_particle() function entails. How this can be extended to Federated learning concept with decentralization nodes: https://doi.org/10.1109/HONET59747.2023.10374608 cross-ref this.
- Provide more implementation details for the multi-threading approach mentioned in Section 3.4 and Algorithm 3. Cross-ref with this https://doi.org/10.1186/s40537-024-00886-w
- In the Evaluation section, include more theoretical analysis and comparisons against other distributed storage solutions beyond just the experimental results.
- Discuss the limitations, assumptions, and potential security/privacy concerns of the proposed system more explicitly.
Reviewer 2 Report
Comments and Suggestions for Authors
The authors in the manuscript entitled “Robust-DSN: A Hybrid Distributed Replication and Encoding Network Ensembled with Distributed Swarm Workflow Scheduler” highlighted limitations of existing network infrastructure processing big data. They presented their implementation based on HYDREN and DSWS. The implementation is encouraging and I appreciate the work. The authors need to consider the following.
1. Abstract lacks a proper flow and require update. It is better to explain the overall operation and then highlight the problems and then move on to the solution presented in the manuscript.
2. Authors have only listed their contributions in Abstract but there is no mention of how these contributions addressed the existing limitations. Conclusions from last paragraph of section 1 can be added to Abstract.
3. Same numerical conclusions to be added to last section “Conclusion”.
4. Related work section is limited. More references should be added in this section. Some references can be moved to this section from other sections.
5. Some abbreviations used without using full text first time.
6. All axes should be same in figures where possible. For example axis “Probability if Data Loss” is ‘0 to 1’ in figure 6 and ‘0 to 0.6’ in figure. Same with other axes.
7. The probability is in % in figure 9 while in previous figures, probability is no in %. Please use a consistent approach.
8. Some references are old. Please use recent references where possible.
Comments on the Quality of English LanguageSome editing is required to improve quality if English.
Reviewer 3 Report
Comments and Suggestions for Authors
The manuscript is an extended version of the previous work (https://ieeexplore.ieee.org/document/10305903).
The topic is interesting and relevant to the journal, but there are issues that prevent publication.
- Awkward Abstract. Contribution is listed in the Abstract. It can be moved to Introduction. Thus, reconstruction of the Abstract is required.
- No main concept or idea has been described in the Abstract.
- No performance summary has been added to the Abstract.
- The authors mentioned "performance, data integrity, and availability". However, no performance evaluation is presented for integrity. Data loss and integrity are different metrics.
- Further, it uses "old" md5. The authors should justify the use of "old" md5.
- For related work, a comparison table with respect to your work can be added for various metrics.
- The evaluation is heavily based on simulations (and probability). No practical implementations or evaluations are provided. For example, Amazon S3 provide 99.99% availability. Why your environment incurs data loss in cloud computing?
- No realistic scenarios are provided for your simulations.
- No complexity of algorithms is provided (it seems to be high). Thus, scheduling time can increase.
- The authors try to hide their experimental parameters and environments. No details are presented.
Comments on the Quality of English LanguageNone
Reviewer 4 Report
Comments and Suggestions for Authors
The manuscript provides a solid overview of the research, its contributions, and the proposed solution.
With some revisions to enhance clarity, coherence, and engagement, it has the potential to make a significant contribution to the field.
1. To increase the impact, consider highlighting the practical implications of the proposed solution and its potential benefits. I believe a note on the practical implications is missing.
2. Section 3.5: Line number 297 and text is Overlapped.
Round 2
Reviewer 3 Report
Comments and Suggestions for Authors
The authors revised the manuscript based on the previous review.
Thus, I recommend the manuscript for publication.
Comments on the Quality of English LanguageNone